# TamperTok: Forensics-Driven Tokenized Autoregressive Framework for Image Tampering Localization

## Abstract

Multi-modal Large Language Models (MLLMs) offer powerful reasoning for localizing tampering in images, yet existing MLLM-based approaches suffer from suboptimal localization due to the reliance on exogenous segmentation decoders. The stitched pipeline introduces information bottlenecks during backpropagation, diluting spatial signals from the MLLM's hidden embeddings and lacking semantic priors for forensic tasks, which leads to imprecise masks and poor generalization in Image Manipulation Detection & Localization (IMDL). To address those limitations, we propose TamperTok, which reformulates MLLM-based IMDL as an autoregressive sequence generation task. Unlike existing approaches relying on exogenous decoder for localization, TamperTok directly generates spatially grounded token sequences from the MLLM, enabling precise probabilistic mask prediction without intermediary supervisions. Specifically, we introduce Kernel Splatting Decoder (KSD) to mitigate the sharp gradients caused by deterministic map in codebook-based detokenizer via clustering-aware code smoothing while mapping tokens to binary masks. In addition, to compensate for the lacking priors of diverse tampering types, *i.e.*, splicing and semantic forgeries, we propose a novel Scene-wise Expert Injection (SwEI) to select and inject multi-scale tampering-specific features from a forensic expert model into the MLLM. Extensive experiments show that TamperTok achieves state-of-the-art (SOTA) performance on multiple tampering localization datasets, with 20% improvements in IoU and F1 over existing MLLM-based models, while exhibiting stronger robustness to noise perturbations and cross-domain scenarios. Codes will be released.

## 1 Introduction

Beyond threats from traditional tampered images, the rapid advancement of AI-generated content (AIGC) has intensified risks like DeepFakes (Mirsky & Lee, 2021), endangering public trust, legal proceedings, and information security. Although prior expert models, such as ManTra-Net (Wu et al., 2019) and SparseViT (Su, 2025), perform well on specific datasets like CASIA (Dong et al., 2009b), they struggle with highly diverse or semantically complex tampering scenarios, as shown by cross-dataset performance gaps (Su, 2025).

Inspired by the powerful and consistent inference of multi-modal large language models (MLLMs), recent methods, such as FakeShield (Xu, 2025) and SIDA (Huang et al., 2025), employ them as reasoning detectors, framing IMDL (image manipulation detection and localization) as a prompt-based task. To enable fine-grained localization, these methods integrate off-the-shelf MLLMs with segmentation decoders (*e.g.*, SAM-like models (Kirillov et al., 2023)), raising two key concerns:

(1) **Ambiguous Representation of Tampering Mask Tokens**. Existing methods generate tampering mask indirectly via external decoders with semantic embeddings from MLLM output. These MLLM embeddings carry the semantic information of the masked region, disregarding the accurate spatial information. The MLLM embeddings may refer to an ambiguous object (*e.g.*, one among many of the same objects) in the images and cannot be precisely supervised by semantic labels. Thus, adapting the existing framework in generating the tampering mask tokens is imprecise.

(2) **Inefficient Fusion of Forensic Knowledge**. Exiting methods like FakeShield and SIDA rely solely on LoRA-based fine-tuning without integrating tampering-specific visual features, limiting their performance in locating fine-grain forensic traces. Moreover, some SOTA approaches that incorporate forensic features through post-backbone multi-scale fusion strategies (*e.g.*, AIGI-Holmes (Zhou et al., 2025)) by simply appending them after the visual encoder. However, the late-stage forensic features and visual features are heterogeneous. Simple fusion of these features without multi-scale alignment is ineffective.

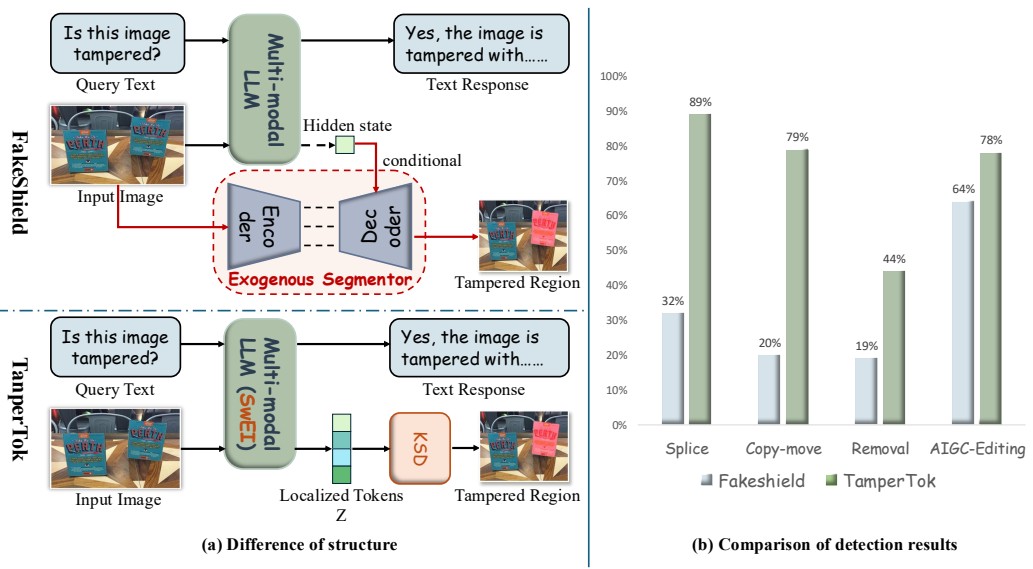

(a) Difference of structure

(b) Comparison of detection results

Figure 1: Comparison of FakeShield (Xu, 2025) and TamperTok. (a) Architectures: FakeShield risks information loss via external decoders (red part); TamperTok uses direct token generation for probabilistic masks. (b) We compared two methods under unified training protocols. F1 scores by tampering type, with TamperTok's average 39% gain over FakeShield.

In this work, we propose an MLLM-based IML model by reformulating the IMDL task as an autoregressive sequence generation task and exploiting discriminative forensic features. To allow precise representation of mask tokens, we introduce our TamperTok framework that directly generate spatially grounded token sequences from the MLLM. It facilitates precise and supervised tampering mask prediction without ambiguous representation. Specifically, inspired by TiTok (Yu et al., 2024), we adopt a pre-trained codebook to map tokens to binary masks, ensuring explicit spatial modeling and end-to-end alignment through coarse to fine decoding. Additionally, to effectively supervise the mask token, we introduce the Kernel Splatting Decoder (KSD) via clustering-aware label smoothing, stabilizing gradients by accounting for semantic similarities in the codebook. To effectively integrate the IMDL-specific knowledge, we introduce the Scene-wise Expert Injection (SwEI) module to inject multi-scale forensic features from expert models into the visual backbone. Specifically, this module aligns features at intermediate layers via attention-based fusion. It preserves subtle cues while mitigating the heterogeneous feature issues of the post-backbone strategies. As shown in Fig. 1, our TamperTok overcomes the existing limitations in mask token representation and IML feature fusion. It achieves consistent spatial reasoning performance with average F1 gains of 0.39 across various tampering types (from FakeShield's 0.33 to 0.72) and robust forensic generalization. Extensive experiments further demonstrate that TamperTok achieves SOTA performance on multiple tampering localization datasets, with 20% improvements in IoU and F1 over existing MLLM-based models, and exhibiting stronger robustness to noise perturbations and cross-domain scenarios.

In summary, our main contributions are:

- We propose a tokenized autoregressive framework for IML, named TamperTok. Our Kernel Splatting Decoder (KSD) allows direct supervision and precise representation of tampering mask tokens.

- We introduce the Scence-wise Expert Injection (SwEI) module to incorporate multi-scale tampering forensic features into the visual encoder in MLLM, enhancing perception of diverse tampering types.

- To the best of our knowledge, this is the first report on MLLM-based model achieving a stronger IML performance than the SOTA forensic experts under challenging cross-domain scenarios.

## 2 RELATED WORKS

### 2.1 IMAGE MANIPULATION DETECTION & LOCALIZATION (IMDL)

IMDL methods aim to generate precise tampering region masks. For deep learning-based IMDL models, the performance improved significantly by data-driven techniques (*e.g.*, ManTra-Net (Wu et al., 2019), MVSS-Net (Chen et al., 2021), and ObjectFormer (Wang et al., 2022)). More recent expert models, such as SparseViT (Su, 2025), refine attention to suspicious regions and achieve strong IMDL performance. Nevertheless, while most forensic methods are lightweight and fast, offering strong perception of low-level information such as texture inconsistencies, they lack broader world knowledge or semantic priors, restricting their ability to handle highly realistic or semantically complex tampering scenarios. In this work, we integrate expert forensic knowledge into MLLMs to provide detailed explanations of forensic decisions.

### 2.2 MULTIMODAL LARGE LANGUAGE MODELS

To adapt existing LLMs for detection and grounding tasks, existing MLLM-based methods (*e.g.*, Pix2Seq (Chen et al., 2022), SEEM (Zou et al., 2023)) align visual features and text prompts via cross-modal mechanisms to generate serialized outputs like masks. These approaches typically extract image features with visual encoders, fuse them with text embeddings via projection layers, and feed the result into decoders for sequence prediction and mask reconstruction. Recent efforts also empower MLLMs for forensic applications, such as explaining synthesized images (He et al., 2025). To allow reasoning-driven segmentation based on the input instruction, LISA (Lai et al., 2024) uses a <seg> token to encode hidden embeddings into masks and facilitate zero-shot generalization for different queries. To boost the performance in complex scenes, HiMTok (Wang et al., 2025) tokenizes visual signals into 1D sequences for MLLM-predicted multi-layer masks.

For MLLM-based IMDL methods, initial attempts such as FakeShield (Xu, 2025) and SIDA (Huang et al., 2025) follow the LISA framework by coupling MLLMs with external decoders (*e.g.*, SAM (Kirillov et al., 2023)) to generate masks from hidden embeddings. However, as highlighted in Sec. 1, these approaches Xu (2025); Huang et al. (2025) suffer from the limitations of ambiguous representation of mask tokens and inefficient fusion of forensic knowledge. To overcome these limitations, we propose a novel TamperTok framework that directly predicts discrete token sequences through MLLMs for mask generation with direct supervision of the Kernel Splatting Decoder (KSD), while incorporating expert forensic features efficiently with Scence-wise Expert Injection (SwEI) module.

## 3 METHODOLOGY

### 3.1 TAMPERTOK

Existing MLLM-based methods (Xu, 2025; Huang et al., 2025) utilize off-the-shelf segmentor (Kirillov et al., 2023) to locate tampered regions with the semantic outputs from MLLM. The stitched design brings architectural overhead and prevents direct supervision from spatial annotations, suffering from information bottlenecks during backpropagation.

To address those problems, we propose a novel framework called TamperTok, which reformulates the localization as detokenizing an autoregressive sequence as shown in Fig. 2. Specifically, we employ **InternVL2.5-8B** (Chen et al., 2024) as the MLLM backbone. To enhance its visual encoder with prior knowledge of tampering, we first introduce the Scene-wise Expert Injection (SwEI). The purpose of SwEI is to take the various advantages of forgery expert according to different tampering methods. Specifically, we propose a multi-scale gate mechanism to fuse the parallel expert features and original visual features adaptively.

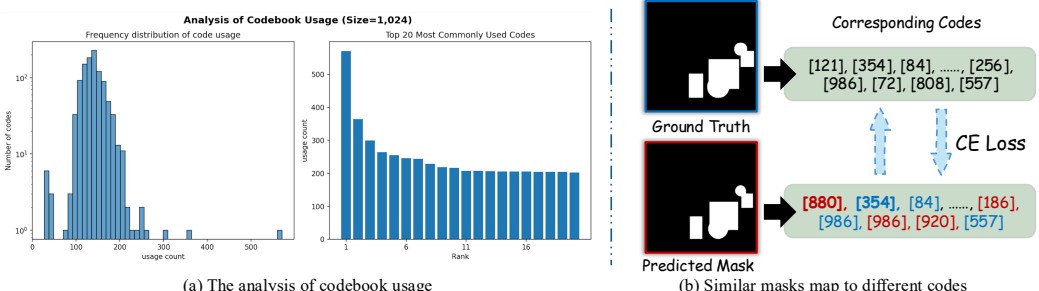

Figure 2: Overview of the proposed TamperTok framework. The architecture integrates the Scene-wise Expert Injection (SwEI) module with the visual backbone, which efficiently incorporates expert forensic knowledge for tampering localization. TamperTok also employs the Kernel Splatting Decoder (KSD), a novel spatial detokenizer that maps tokens to binary masks.

As for the location, we propose a novel segmentation paradigm, the core design of which is Kernel Splatting Decoder (KSD). To be specific, KSD is a spatial detokenizer to map the tokens to binary mask directly. Following the Large Language Model (LLM), KSD decodes a series of output tokens to mask. Inspired by (Wang et al., 2025; Yu et al., 2024), KSD will convert tokens into tampering probabilities for different regions of the mask based on a codebook, which is pretrained on binary masks. These tokens will be clustered into several similar groups in KSD to alleviate the sharp gradients caused by hard decoding for better proformance.

## 3.2 KERNEL SPLATTING DECODER

Figure 3: Analysis of codebook usage and the sharp gradient problem. (a) The usage of codes is well-balanced, with no redundancy in the codes. (b) However, vastly different code sequences could be decoded to similar masks, leading to sharp gradients during the optimization of the cross-entropy loss.

The original MLLM architecture is suitable for autoregressive output of token sequences but is powerless when it comes to localizing 2D masks. To track the tampered regions, existing MLLM-based methods bring in an external segmentor for fine-grained masks, while only a few semantic tokens are used as the link. The indirect references of semantic tokens fail to fully leverage the capabilities of the segmentor, often resulting in unsatisfied localization performance. HiMTok (Wang et al., 2025) brings a novel perspective to design a codebook-based detokenizer, which maps a fixed number of tokens to mask, which can be formulated as:

$$\hat{z} = \operatorname{argmin}_k \|z - e_k\|_2, \tag{1}$$

while the $z$ is the token feature and $e_k$ means the $k$-th feature in codebook. A serise of $\hat{z}$ will be map to a 2D mask. However, HiMTok has a significant limitation. The combination of Eq. 1 and cross-entropy supervision brings sharp gradient in backpropagation. Assume there are two encoded

features are close, $e_1 \approx e_2$, the hard assignment may map $z$ to different codes like $\hat{z} = e_1$ and $\hat{z} = e_2$, resulting in large differences in the cross-entropy loss:

$$\Delta \mathcal{L}_{\text{CE}} = \mathcal{L}_{\text{CE}}(z, e_1) - \mathcal{L}_{\text{CE}}(z, e_2) \gg 0, \tag{2}$$

and consequently, drastic gradient variations:

$$\|\nabla_\theta \mathcal{L}_{\text{CE}}(z, e_1) - \nabla_\theta \mathcal{L}_{\text{CE}}(z, e_2)\| \gg \|e_1 - e_2\|. \tag{3}$$

At the same time, we observe, as shown in Fig. 3, that although all codes are used equally during the decoding process, some codes exhibit significant feature similarity. Based on that, we propose Kernel Splatting Decoder (KSD), which leverages the similarity of codes in the codebook to mitigate the harmful effects of sharp gradients. Specifically, during the training process, all codes in the codebook will be clustered based on their similarity. We first cluster the codes in the codebook based on their similarity, and reformulate the original process, which maps features to fixed code, as decoding to the nearest cluster instead. The Eq. 1 will be reformulated as

$$\hat{z}_{\text{cluster}} = \arg\min_{\mu_m} \|z - \mu_m\|_2, \tag{4}$$

in which:

$$\mu_m = \frac{1}{|\mathcal{G}_m|} \sum_{e_k \in \mathcal{G}_m} e_k, \tag{5}$$

where $\mathcal{G}_m$ is the $m$-th cluster. After identifying the corresponding cluster, the query token computes a normalized weight based on the similarity to the codes within the cluster, and applies it to the cross-entropy loss of the corresponding codes as:

$$L_{KSD} = \sum_{i=1}^{K} \underbrace{\frac{\exp\left(\frac{z \cdot e^i}{\sqrt{d}}\right)/\tau}{\sum_{j=1}^{K} \exp\left(\frac{z \cdot e^j}{\sqrt{d}}\right)/\tau}}_{\text{weight from similarity}} \cdot \underbrace{L_{CE}^i}_{\text{CE loss of code } i}. \tag{6}$$

Let's review the assumed closed code $e_1$ and $e_2$. When we replace the original cross-entropy loss to Eq. 6, the new gradient them would be:

$$\frac{\partial L}{\partial z} = \sum_{i=1}^{K} w_i \frac{\partial L_i}{\partial z} + \frac{1}{\sqrt{d}} \sum_{i=1}^{K} w_i L_i (e_i - \mu), \tag{7}$$

where $w_i$ is the weight of $i$-th code in cluster and $\mu$ is the center of the cluster. It is evident that the gradient of KSD is smoother, ensuring stable convergence during training.

### 3.3 SCENE-WISE EXPERT INJECTION

While MLLM excels at semantic reasoning, it lacks the task-related knowledge, which is the culprit of poor generalization in downstream scene. A few recent methods try to fuse the features from experts and original visual features. However, we noticed that for IMDL, varied tampering techniques challenges the simple thought. Relying directly on the experts yields mediocre results even after fine-tuning on tampering data.

To bridge this gap, our Scene-wise Expert Injection (SwEI) is designed to selectively inject multi-scale tampering forensic features into the visual encoder in MLLM, enhancing perception of diverse tampering types. For instance, SwEI adaptively fuse low-level features for splicing boundaries or high-level semantics for copy-move duplications.

In details, we introduce SparseViT (Su, 2025), which is the recent SOTA and has the similar structure compared with the visual encoder, as the expert. With an image $\boldsymbol{I}$, we first use expert to extract two progressive features $\boldsymbol{E}_s$ and $\boldsymbol{E}_d$, while the front is from the shallow layer and the later is from deeper one. For the $i$-th stage, the features from visual backbone are be formulated as $\boldsymbol{F}^i$. We first want make all the features shape same:

$$\tilde{\boldsymbol{E}}_s = \text{resize}(\boldsymbol{E}_s, \text{size}(\boldsymbol{F}^i)), \quad \tilde{\boldsymbol{E}}_d = \text{resize}(\boldsymbol{E}_d, \text{size}(\boldsymbol{F}^i)), \tag{8}$$

while size($\cdot$) means extract the shape of feature. After that, we compute the attention weight of two expert features as:

$$[\alpha^i, \beta^i] = \text{Softmax} \left( \frac{(W_q^i \boldsymbol{F}^i)^\top [W_s^i \tilde{\boldsymbol{E}}_s, W_d^i \tilde{\boldsymbol{E}}_d]}{\sqrt{d}} \right),$$

(9)

where $\alpha^i$ and $\beta^i$ mean the normalized weight of various expert information, and $W$ is the map matrix related to the according features. Then we integrate all the expert information with the weight to form the sparse-enhanced feature, which can be described as:

$$\hat{\boldsymbol{F}}^i = \boldsymbol{F}^i + \text{FFN}(\alpha^i \tilde{\boldsymbol{E}}_s + \beta^i \tilde{\boldsymbol{E}}_d),$$

(10)

where $\hat{\boldsymbol{F}}^i$ continues to forward pass through the original encoder structure and is ultimately being fed into the LLM with the textual features.

Through the SwEI process outlined above, we are able to selectively inject multi-scale expert information at each stage of the visual encoder. SwEI further deconstructs the expert internally, enabling the new feature $\hat{\boldsymbol{F}}^i$ to more flexibly address various types of tampering. For instance, in the case of mosaic tampering, which disrupts image pixels, SwEI places greater emphasis on capturing low-level textures. In contrast, for copy-move tampering, which has fewer traces of manipulation, SwEI emphasizes the extraction of disharmonious semantics. The final loss is a combination of the LLM supervision loss, the KSD loss, and the localization loss.

## 4 EXPERIMENTS

We evaluate our model on benchmark datasets for image tampering localization, starting with setup (datasets, details), then quantitative results and analysis.

### 4.1 EXPERIMENTAL SETUP

#### 4.1.1 DATASETS AND EVALUATION METRICS

To ensure a fair and comprehensive evaluation, we design an experimental protocol that emphasizes cross-domain generalization for MLLMs, differing from protocols tailored for specialized forensic models (*e.g.*, SparseViT (Su, 2025)), which often neglect AIGC-based manipulations, and from in-domain evaluations in SIDA (Huang et al., 2025). Our protocol incorporates a broader mix of datasets to enhance reasoning capabilities and robustness across diverse tampering scenarios.

Our training process utilizes TamperCOCO (Kwon et al., 2021b), MIML (Qu et al., 2024), CA-SIA2 (Dong et al., 2009b), and tampered samples from SID_Set (Huang et al., 2025). These datasets—TamperCOCO, MIML, and CASIA2—are commonly used in IMDL research (Ma et al., 2024; Qu et al., 2024; Nandi et al., 2023). Additionally, SID_Set, focused on AIGC tampering, strengthens the model's detection of sophisticated AI-generated forgeries.

Evaluation is conducted on six diverse benchmarks to assess cross-domain localization and generalization on unseen datasets: CASIA1 (Dong et al., 2009a), NIST (Yates et al., 2017), Coverage (Cozzolino et al., 2016), Columbia (Ng et al., 2004), Glide (Guillaro et al., 2023), and IMD (Novoz'amsk'y et al., 2020). These encompass various tampering operations (*i.e.*, splicing, copy-move, removal, and AIGC-based forgeries).

Following prior IML literature, we emphasize pixel-level metrics: Intersection over Union (IoU) and F1 score, which measure overlap and accuracy between predicted and ground-truth tampering masks.

### 4.2 LOCALIZATION PERFORMANCE RESULTS

Table 1 presents a comprehensive comparison on the tampering localization performance of our TamperTok against both forensic expert models (*e.g.*, CAT-Net (Kwon et al., 2021a), MVSS-Net (Chen et al., 2021), PSCC-Net(Liu, 2022), TruFor(Guillaro et al., 2023), SparseViT (Su, 2025)) and MLLM-based methods (*e.g.*, FakeShield (Xu, 2025), SIDA (Huang et al., 2025)). All compared

methods were trained on the same datasets outlined in Sec. 4.1.1. TamperTok achieves the highest average IoU (0.67) and F1 (0.79) scores, surpassing all existing methods across most datasets.

Compared to forensic expert models, TamperTok outperforms the top baseline SparseViT (average IoU 0.64, F1 0.76) by 0.03 in IoU and 0.03 in F1. The gains are particularly pronounced on CASIA1 (+0.11 IoU, +0.05 F1). This is because the CASIA1 dataset is dominated by splicing forgeries with semantically distinct manipulations. In these samples, our MLLM-based approach leverages semantic reasoning capabilities to achieve good tampering localization performance. In contrast, our gains on Coverage are marginal since this dataset emphasis on copy-move forgeries. Detecting such samples mainly rely on non-semantic, fine-grained features like texture inconsistencies, and forensic experts already excel in these scenarios with data-driven features.

Against MLLM-based methods, TamperTok significantly outpaces SIDA (average IoU 0.44, F1 0.57) by 0.23 in IoU and 0.22 in F1, and FakeShield (0.35/0.46) by 0.32/0.33. These substantial improvements mainly arise from our contribution on reformulation of IMDL as autoregressive token generation problem. It allows direct supervision of the MLLM outputed token, and eliminates information bottlenecks and semantic mismatches from exogenous segmentation decoders.

Visual comparisons in Fig. 4 further highlight TamperTok's advantages. Our model not only accurately localizes tampered regions while effectively excluding interference from non-tampered areas but also generates clear and complete masks across diverse scenarios. For instance, when compared to the forensic expert model SparseViT, both methods successfully identify the primary tampered regions; however, SparseViT often yields false alarm in pristine pixels. In contrast, against MLLM-based methods like SIDA and FakeShield, which either fail to pinpoint accurate tampered regions or produce blurry segmentation, TamperTok delivers precise and complete localization regardless of whether the tampering involves obvious semantic inconsistencies or subtle non-semantic cues.

Table 1: Comparison of tampering localization performance for TamperTok against traditional forensic methods and state-of-the-art MLLM-based approaches, evaluated using IoU and F1 metrics across six datasets. Underlined values highlight the second-best scores among Forensic MLLM methods, while the red percentages indicate TamperTok's improvement margins relative to these second-best scores.

| Method | CASIA1 | | NIST | | Coverage | | Columbia | | Glide | | IMD | | Average | |
|---|---|---|---|---|---|---|---|---|---|---|---|---|---|---|
| | IOU | F1 | IOU | F1 | IOU | F1 | IOU | F1 | IOU | F1 | IOU | F1 | IOU | F1 |
| *Forensic Expert* | | | | | | | | | | | | | | |
| CAT-Net | 0.38 | 0.56 | 0.29 | 0.33 | 0.13 | 0.23 | 0.77 | 0.77 | 0.28 | 0.43 | 0.30 | 0.34 | 0.36 | 0.44 |
| MVSS-Net | 0.49 | 0.70 | 0.25 | 0.30 | 0.43 | 0.59 | 0.86 | 0.90 | 0.44 | 0.62 | 0.40 | 0.53 | 0.48 | 0.61 |
| PSCC-Net | 0.29 | 0.43 | 0.20 | 0.26 | 0.30 | 0.43 | 0.38 | 0.54 | 0.39 | 0.58 | 0.17 | 0.27 | 0.29 | 0.42 |
| TruFor | 0.60 | 0.74 | 0.36 | 0.43 | 0.46 | 0.50 | 0.75 | 0.78 | 0.48 | 0.66 | 0.48 | 0.59 | 0.52 | 0.62 |
| SparseViT | 0.68 | 0.82 | 0.39 | 0.57 | **0.62** | **0.79** | **0.89** | 0.88 | 0.65 | 0.77 | 0.60 | 0.73 | 0.64 | 0.76 |
| *Forensic MLLM* | | | | | | | | | | | | | | |
| Fakeshield | 0.48 | 0.68 | 0.17 | 0.22 | 0.27 | 0.46 | 0.68 | 0.75 | 0.47 | 0.64 | 0.02 | 0.03 | 0.35 | 0.46 |
| SIDA | 0.46 | 0.63 | 0.27 | 0.43 | 0.30 | 0.45 | 0.65 | 0.70 | 0.48 | 0.58 | 0.46 | 0.61 | 0.44 | 0.57 |
| TamperTok | 0.79↑31% | 0.87↑19% | 0.43↑16% | 0.59↑16% | 0.61↑31% | 0.79↑33% | 0.89↑21% | 0.90↑15% | 0.66↑18% | 0.78↑14% | 0.65↑19% | 0.78↑17% | 0.67↑23% | 0.79↑22% |

## 4.3 ABLATION STUDY

### 4.3.1 ARCHITECTURE ABLATION

As shown in Tab. 2, we conduct an ablation study on the key components in our model. Note that the baseline without any of the three components (*i.e.*, the first row in the table) corresponds to the vanilla LISA design Lai et al. (2024).

**Codebook-based Decoder Ablation**: Our variant with only the codebook-based decoder (No.2) improves performance to an average F1 of 0.74, featuring a 0.31 gain over the LISA baseline (No.1). It highlights the benefits of autoregressive tokens with codebook-based decoder.

**Expert Injection Ablation**: The variant with SwEI (No.3) shows a performance gain of 0.03 in average F1 score over that of the second variant. Coverage benefits the most (+0.13) among all datasets, emphasizing the role of SwEI in detecting subtle copy-move inconsistencies.

**Kernel Splatting Decoder Ablation**: The variant with kernel splatting (No.4) demonstrate 0.02 gain in F1 score over that of the third variant. It confirms the benefits of clustering-aware smoothing for stabilizing token gradients.

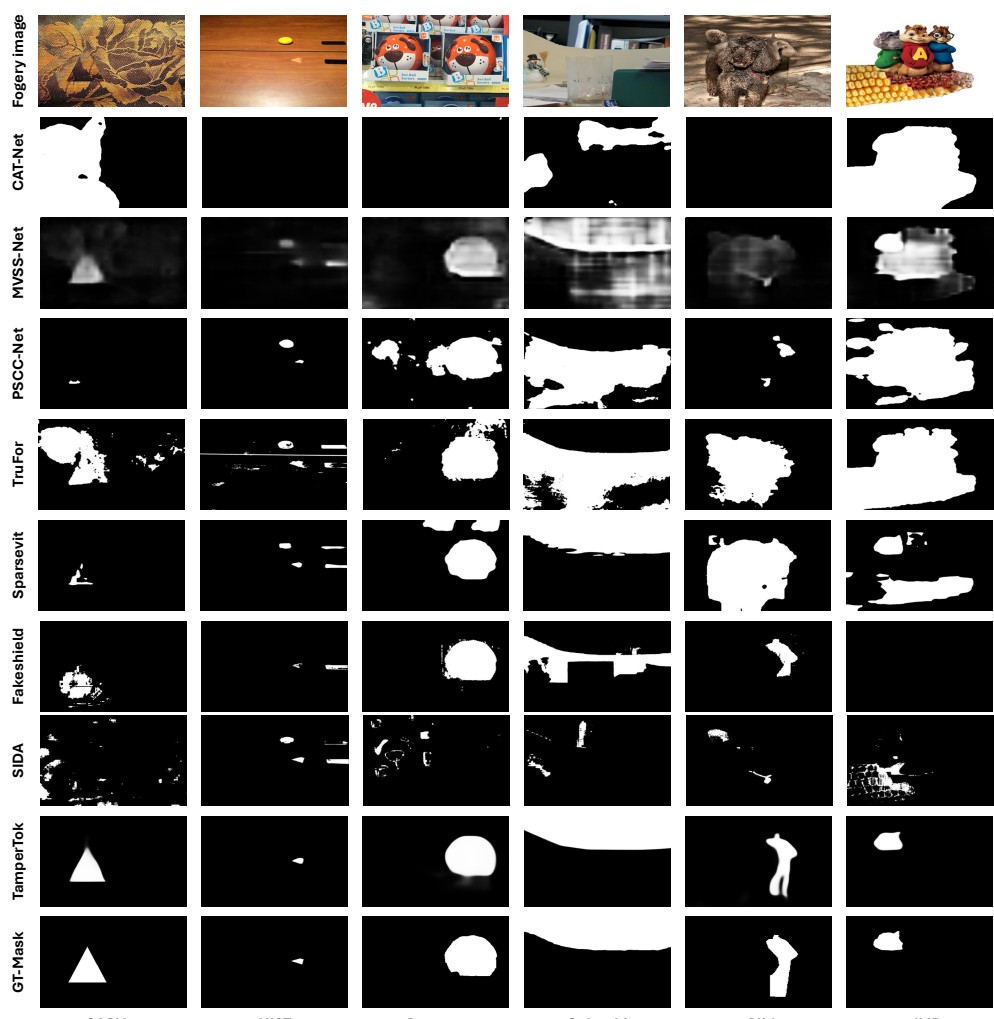

Figure 4: Qualitative comparisons between our TamperTok and other competitive methods.

Table 2: Ablation study for TamperTok with **IOU** and **F1** across six datasets."CBD" = *Codebook-based Decoder*, "SwEI" = *Scene-wise Expert Injection*, "KSD" = *Kernel Splatting Decoder*.The configuration with none of the three components corresponds to the LISA baseline.

| CBD | SwEI | KSD | CASIA1 | | NIST | | Coverage | | Columbia | | Glide | | IMD | | Average | |
|-----|------|-----|--------|------|------|------|----------|------|----------|------|-------|------|------|------|---------|------|
| | | | IOU | F1 | IOU | F1 | IOU | F1 | IOU | F1 | IOU | F1 | IOU | F1 | IOU | F1 |
| ✗ | ✗ | ✗ | 0.30 | 0.48 | 0.29 | 0.37 | 0.19 | 0.33 | 0.39 | 0.55 | 0.28 | 0.44 | 0.30 | 0.40 | 0.29 | 0.43 |
| ✓ | ✗ | ✗ | 0.73 | 0.86 | 0.40 | 0.56 | 0.47 | 0.66 | 0.86 | 0.85 | 0.61 | 0.79 | 0.60 | 0.73 | 0.61 | 0.74 |
| ✓ | ✓ | ✗ | 0.75 | 0.87 | 0.38 | 0.56 | 0.57 | 0.74 | 0.88 | 0.88 | 0.65 | **0.80** | 0.62 | 0.75 | 0.64 | 0.77 |
| ✓ | ✓ | ✓ | **0.79** | **0.87** | **0.43** | **0.59** | **0.61** | **0.79** | **0.89** | **0.90** | **0.66** | 0.78 | **0.65** | **0.78** | **0.67** | **0.79** |

### 4.3.2 FEATURE FUSION ABLATION

To demonstrate the superiority of our intermediate fusion strategy in the SwEI, we compare it against post-backbone fusion strategies, with the SparseViT branch fixed. For sequence concatenation, we adopt the approach from (Zhou et al., 2025), which concatenates visual features token-wise, resulting in $2N$ tokens fed to the LLM after projection. For dimension concatenation, we implement

a baseline variant that appends sparse features channel-wise followed by a $2H$ to $H$ projection before feeding to the LLM.

As shown in Tab. 3, our intermediate fusion achieves an average IoU of 0.67 and F1 of 0.79, outperforming the performances of other fusion strategies. It is because both sequence and dimension concatenation are late-stage strategies, which dilute forensic cues through suboptimal integration between heterogeneous features. In contrast, our intermediate fusion preserves multi-scale alignment, yielding notable gains on CASIA1 (+0.07 IoU vs. dimension concatenation) and Coverage (+0.09 IoU vs. dimension concatenation).

Table 3: Performance comparisons of different feature fusion strategy in IoU and F1 scores.

| Fusion Method | CASIA1 | | NIST | | Coverage | | Columbia | | Glide | | IMD | | Average | |
|---|---|---|---|---|---|---|---|---|---|---|---|---|---|---|
| | IOU | F1 | IOU | F1 | IOU | F1 | IOU | F1 | IOU | F1 | IOU | F1 | IOU | F1 |
| Dimension Concat | 0.72 | 0.86 | 0.33 | 0.52 | 0.52 | 0.69 | 0.89 | 0.89 | 0.59 | 0.75 | 0.61 | 0.75 | 0.61 | 0.74 |
| Sequence Concat | 0.69 | 0.80 | 0.34 | 0.53 | 0.57 | 0.71 | 0.88 | 0.87 | 0.64 | 0.79 | 0.59 | 0.73 | 0.62 | 0.74 |
| SwEI | **0.79** | **0.87** | **0.43** | **0.59** | **0.61** | **0.79** | **0.89** | **0.90** | **0.66** | **0.78** | **0.65** | **0.78** | **0.67** | **0.79** |

### 4.3.3 KSD CLUSTERING HYPER-PARAMETER ABLATION

To assess the sensitivity of the Kernel Splatting Decoder (KSD) to the clustering hyper-parameter $K$ (`smoothing_topk`), we conducted an ablation study with varying $K$ values. The results are summarized in Table 4. As observed, setting $K = 1$ (equivalent to hard decoding) leads to sharp gradients and suboptimal convergence, resulting in an Average F1 of 0.77. Increasing $K$ to 10 provides effective label smoothing, yielding the best overall performance (Average F1: 0.79). However, further increasing $K$ to 20 results in over-smoothing, which dilutes the distinctive spatial features of the masks and degrades performance to 0.76. Therefore, we adopt $K = 10$ as the optimal setting.

Table 4: Ablation study on the KSD clustering hyper-parameter $K$. We report IoU and F1 scores across six datasets. $K = 10$ yields the best trade-off between gradient smoothing and feature distinctiveness.

| $K$ | CASIA1 | | NIST | | Coverage | | Columbia | | Glide | | IMD | | Average | |
|---|---|---|---|---|---|---|---|---|---|---|---|---|---|---|
| | IoU | F1 | IoU | F1 | IoU | F1 | IoU | F1 | IoU | F1 | IoU | F1 | IoU | F1 |
| $K = 1$ (Hard) | 0.75 | 0.87 | 0.38 | 0.56 | 0.57 | 0.74 | 0.88 | 0.88 | 0.65 | 0.80 | 0.62 | 0.75 | 0.64 | 0.77 |
| $K = 5$ | 0.77 | 0.85 | 0.37 | 0.54 | 0.60 | 0.74 | **0.89** | 0.88 | 0.65 | **0.79** | 0.63 | 0.75 | 0.65 | 0.76 |
| $K = 10$ (**Ours**) | **0.79** | **0.87** | **0.43** | **0.60** | **0.61** | **0.79** | **0.89** | **0.90** | **0.66** | 0.78 | **0.65** | **0.78** | **0.67** | **0.79** |
| $K = 20$ | 0.77 | **0.87** | 0.36 | 0.55 | 0.57 | 0.73 | **0.89** | 0.88 | 0.63 | 0.76 | 0.63 | 0.76 | 0.64 | 0.76 |

### 4.3.4 EXPERT MODEL SELECTION ABLATION

We further investigated the rationale for selecting SparseViT as the forensic expert by replacing it with other expert models (MVSS-Net and PSCC-Net) within the SwEI module. The results in Table 5 demonstrate that while incorporating any of these expert models consistently brings performance improvements to the framework, SparseViT achieves the best absolute performance (0.79 F1). Furthermore, given that the differences in computational burden among these lightweight expert models are negligible relative to the substantial overhead of the MLLM backbone, we finally adopted SparseViT as the optimal choice based on its superior performance.

## 4.4 ROBUSTNESS STUDY

To evaluate robustness against real-world degradations (Wu et al., 2022; Bayar & Stamm, 2018), we apply JPEG compression (quality levels of 80% and 70%), Gaussian noise (variance $\sigma^2 = 5$ and 10), and resizing distortions (factor of 0.75 and 0.5) to test images across all datasets. To keep the original resolution under resizing, we up-sample the images with corresponding scaling factor using bicubic interpolation. We compare TamperTok with FakeShield (Xu, 2025) and SIDA (Huang et al., 2025) by reporting average F1 scores under these perturbations.

Table 5: Ablation study on the choice of expert models within the SwEI module. We compare the performance of standalone experts (Rows 1, 3, 5) against their integration into TamperTok (Rows 2, 4, 6). The results indicate that our framework consistently enhances the capabilities of various experts, with SparseViT yielding the optimal performance.

| Method | CASIA1 | | NIST | | Coverage | | Columbia | | Glide | | IMD | | Average | |
|--------|--------|-----|------|-----|----------|-----|----------|-----|-------|-----|-----|-----|---------|-----|
| | IoU | F1 | IoU | F1 | IoU | F1 | IoU | F1 | IoU | F1 | IoU | F1 | IoU | F1 |
| *Expert Only:* MVSS-Net | 0.49 | 0.70 | 0.25 | 0.30 | 0.43 | 0.59 | 0.86 | 0.90 | 0.44 | 0.62 | 0.40 | 0.53 | 0.48 | 0.61 |
| **TamperTok w/ MVSS** | 0.70 | 0.84 | 0.31 | 0.35 | 0.39 | 0.53 | 0.89 | 0.89 | 0.56 | 0.74 | 0.58 | 0.71 | 0.57 | 0.68 |
| *Expert Only:* PSCC-Net | 0.29 | 0.43 | 0.20 | 0.26 | 0.30 | 0.43 | 0.38 | 0.54 | 0.39 | 0.58 | 0.17 | 0.27 | 0.29 | 0.42 |
| **TamperTok w/ PSCC** | 0.73 | 0.86 | 0.39 | 0.53 | 0.45 | 0.58 | 0.86 | 0.86 | 0.59 | 0.75 | 0.59 | 0.71 | 0.60 | 0.72 |
| *Expert Only:* SparseViT | 0.68 | 0.82 | 0.39 | 0.57 | **0.62** | **0.79** | **0.89** | 0.88 | 0.65 | 0.77 | 0.60 | 0.73 | 0.64 | 0.76 |
| **TamperTok w/ SparseViT** | **0.79** | **0.87** | **0.43** | **0.60** | 0.61 | **0.79** | **0.89** | **0.90** | **0.66** | **0.78** | **0.65** | **0.78** | **0.67** | **0.79** |

Table 6: Robustness comparison of different approaches in average F1 scores.

| Method | Original | JPEG-80% | JPEG-70% | Gauss ($\sigma^2 = 10$) | Gauss ($\sigma^2 = 5$) | Resize 0.5 | Resize 0.75 |
|--------|----------|----------|----------|---------------------------|--------------------------|------------|-------------|
| FakeShield | 0.46 | 0.42 | 0.41 | 0.44 | 0.45 | 0.42 | 0.43 |
| SIDA | 0.57 | 0.54 | 0.54 | 0.51 | 0.54 | 0.54 | 0.53 |
| TamperTok | **0.79** | **0.68** | **0.66** | **0.71** | **0.73** | **0.67** | **0.73** |

As shown in Tab. 6, TamperTok consistently outperforms the baselines across all perturbations. It exhibits strongest robustness under resize-0.75 (F1 of 0.73), surpassing FakeShield (0.43) by 0.30 and SIDA (0.53) by 0.20, owing to SwEI's integration of multi-scale forensic features that preserve subtle cues like texture inconsistencies during moderate downsampling—unlike baselines, which rely on distortion-tolerant but less precise high-level semantics. Conversely, TamperTok shows relative weakness under JPEG-70% (F1 of 0.66), exceeding FakeShield (0.41) by 0.25 and SIDA (0.54) by 0.12, as strong compression blurs edge artifacts critical to its scene-wise expert injection. Baselines like SIDA and FakeShield experience minimal degradation (*e.g.*, 0.03–0.06 F1 drops for SIDA), reflecting their semantic focus, whereas TamperTok's superior resilience arises from autoregressive token generation that sustains fine-grained tampering cues under degradation.

## 5 CONCLUSION

This work propose an MLLM-based IML framework, TamperTok, by reformulating the IMDL task as an autoregressive sequence generation task. TamperTok features with KSD that allow precise representation of tampering mask tokens, and SwEI module that incorporates discriminative forensic features. In the experiment, we compare TamperTok with the SOTA forensic experts under challenging cross-domain protocols. Our results demonstrate that TamperTok achieving stronger IML performance than the SOTA forensic experts. In the future, we plan to investigate the explainable text output of TamperTok for the forensic decision process.

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

## A APPENDIX

### A.1 USE OF LLMS

In the preparation of this paper, large language models (LLMs) were used solely to aid and polish the writing. Specifically, they assisted with tasks such as suggesting alternative phrasing for clarity, correcting grammatical errors, improving sentence structure for better flow, and ensuring consistency in terminology and style. For example, LLMs were prompted to reword awkward sentences or

provide synonyms to enhance readability, but all original content, ideas, and technical details were generated by the authors. They were not employed for retrieval and discovery (*e.g.*, finding related work), for research ideation, or for generating any substantive scientific content, such as hypotheses, methodologies, or conclusions.

## A.2 IMPLEMENTATION DETAILS

We initialize the model with pre-trained weights from InternVL2.5-8B (Chen et al., 2024) for the MLLM backbone, HiMTok (Wang et al., 2025) for the decoder, and SparseViT (Su, 2025) for expert features, fully fine-tuning all parameters. Specifically, for the decoder, we utilize the codebook weights inherited from HiMTok. The codebook was pre-trained on a large-scale composite segmentation dataset, including ADE20K, COCO, etc. To adapt the codebook to tampering localization, we fine-tune it using the Hierarchical Mask Loss (HML) alongside our proposed Kernel Splatting Decoder loss ($L_{KSD}$). We use AdamW optimizer with a base learning rate of 2e-5 (increased to 4e-5 for the Scene-wise Expert Injection (SwEI) module to accelerate adaptation), weight decay of 0.05. Images are resized to $448\times448$ resolution with random flipping and color jittering for augmentation, alongside dynamic image sizing, thumbnail usage, and data resampling.

## A.3 DETECTION PERFORMANCE RESULTS

We adhere to the experimental settings outlined in the SIDA paper (Huang et al., 2025) for evaluations on the SID-Set dataset. The dataset is partitioned into training, validation, and test sets in a 7:1:2 ratio. All models presented in the table are trained on the SID-Set training split and evaluated on the test set. In line with the detection metrics employed in (Huang et al., 2025), we report per-class Accuracy (ACC) and F1 scores, along with their macro-averaged values across the three classes, denoted as Overall." For classification, the SIDA model directly outputs category predictions, whereas all other models derive predictions from their output masks: a fully background mask indicates Real," a fully foreground mask indicates Fully Synthetic," and any intermediate mask indicates Tampered."

Table 7: Classification performance on **SID-Set** using the SIDA protocol. We report per-class Accuracy (ACC) and F1, and their macro averages (Overall).

| Method | Real | | Fully Synthetic | | Tampered | | Overall | |
| --- | --- | --- | --- | --- | --- | --- | --- | --- |
| | ACC | F1 | ACC | F1 | ACC | F1 | ACC | F1 |
| *Forensic Expert* | | | | | | | | |
| MVSS | 0.98 | 0.98 | 0.97 | 0.96 | 0.97 | 0.95 | **0.97** | 0.97 |
| PSCC | **0.99** | **0.99** | 0.36 | 0.53 | 0.01 | 0.03 | 0.45 | 0.52 |
| TruFor | 0.73 | 0.84 | **0.99** | **0.99** | **0.98** | **0.99** | 0.90 | 0.94 |
| SparseViT | 0.96 | 0.98 | **0.99** | **0.99** | 0.93 | 0.96 | 0.96 | **0.98** |
| *Forensic MLLM* | | | | | | | | |
| SIDA | 0.89 | 0.91 | 0.98 | 0.98 | 0.92 | 0.91 | 0.93 | 0.93 |
| TamperTok | **0.99** | 0.97 | **0.99** | **0.99** | 0.94 | 0.97 | **0.97** | **0.98** |

As shown in Table 7, TamperTok attains superior overall performance, with macro-averaged ACC and F1 scores of 0.97 and 0.98, respectively, surpassing SIDA (0.93/0.93) and matching or exceeding forensic expert models such as SparseViT (0.96/0.98). Compared to forensic expert models, TamperTok demonstrates competitive or improved performance across categories, particularly in the Real" class (ACC: 0.99, F1: 0.97), underscoring its robustness in distinguishing authentic images. Relative to SIDA, TamperTok exhibits notable gains in all classes, including a 0.10/0.06 improvement in ACC/F1 for the Real" class and a 0.02/0.06 enhancement for the "Tampered" class. These advancements stem from TamperTok's reformulation of IMDL as an autoregressive sequence generation task, which directly generates spatially grounded token sequences from the MLLM to enable precise probabilistic mask prediction without intermediary supervisions, while incorporating

tampering-specific visual features via the Scene-wise Expert Injection (SwEI) module to address the limitations of inefficient fusion of forensic knowledge.

### A.4 ANALYSIS OF ADAPTIVE EXPERT FEATURE INJECTION

To investigate how the Scene-wise Expert Injection (SwEI) module adapts to different manipulations, we define the *Average Relative Expert Intensity* ($\mathcal{I}_{avg}$) to quantify the magnitude ratio of auxiliary expert features relative to backbone features. The calculation involves the following formulations:

$$\mathcal{I}_{avg} = \frac{1}{B} \sum_{i=1}^{B} \frac{\text{RMS}(\mathbf{F}_{\text{sparse}}^{(i)})}{\text{RMS}(\mathbf{F}_{\text{vit}}^{(i)})} \tag{11}$$

$$\text{RMS}(\mathbf{X}) = \sqrt{\frac{1}{C \cdot H \cdot W} \sum_{c,h,w} \mathbf{X}_{c,h,w}^2} \tag{12}$$

$$\mathbf{F}_{\text{sparse}} = \alpha \cdot \text{Proj}(\tilde{\boldsymbol{E}}_s) + \beta \cdot \text{Proj}(\tilde{\boldsymbol{E}}_d) \tag{13}$$

where $B$ denotes the number of samples in the evaluation set. $\text{RMS}(\mathbf{X})$ is the Root Mean Square function calculating the quadratic mean magnitude over the channel ($C$), height ($H$), and width ($W$) dimensions. $\mathbf{F}_{\text{vit}}$ represents the intermediate feature map extracted from the MLLM backbone (InternVL) at the corresponding layer, while $\mathbf{F}_{\text{sparse}}$ is the adaptively fused expert feature map derived from the Scene-wise Expert Injection (SwEI) module. $\tilde{\boldsymbol{E}}_s$ and $\tilde{\boldsymbol{E}}_d$ denote the multi-scale feature maps with complementary characteristics (e.g., texture-rich and semantic-rich) from the expert model (SparseViT). $\text{Proj}(\cdot)$ is the projection layer that aligns the channel dimensions of expert features with the backbone. Finally, $\alpha$ and $\beta$ refer to the dynamic attention maps generated by the backbone features, used to adaptively **weight** the contributions of different expert scales.

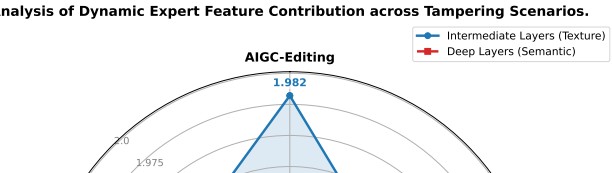
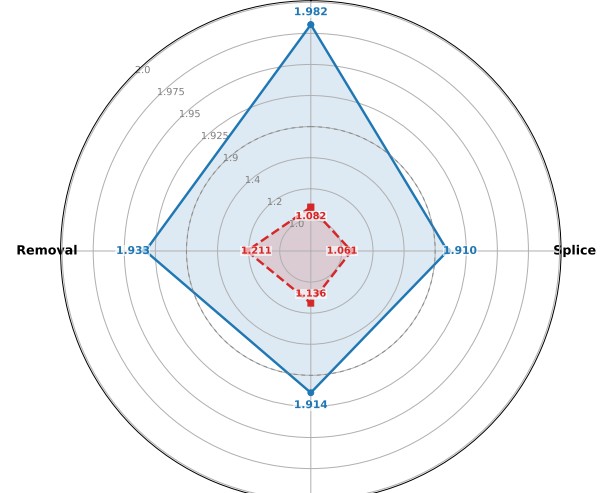

Figure 5: Visualization of Adaptive Expert Feature Injection Intensity ($\mathcal{I}_{avg}$). The SwEI module dynamically modulates injection: **intermediate layers** peak for texture-rich AIGC forgeries, while **deep layers** peak for structure-dependent Removal tampering.

As shown in Figure 5, SwEI acts as a dynamic compensator rather than a static addition:

- **AIGC-Editing (Texture-Driven):** The expert intensity peaks at 1.982 in the **intermediate layers**. This confirms that the model actively prioritizes expert injection to capture subtle

low-level artifacts, such as diffusion fingerprints. It effectively compensates for the MLLM backbone, which typically lacks sensitivity to high-frequency details.

- **Removal (Semantic-Driven):** We observe the highest **deep layer** intensity (1.211) for removal operations. This indicates a necessity for expert guidance to identify "inpainted" areas. Although these regions often fit the semantic context, they introduce structural inconsistencies that the backbone alone may fail to detect.

- **Splicing & Copy-Move:** Splicing exhibits the lowest deep-layer intensity (1.061), suggesting the MLLM backbone is sufficient for **detecting** blatant context contradictions like object mismatches. Conversely, Copy-Move requires higher expert engagement (1.136) in the deep layers, utilizing structural cues to detect repeated internal patterns.

### A.5 EXPLAINABILITY FINE-TUNING EXPERIMENTS

To enhance the explainability of TamperTok, we performed fine-tuning using the ForgeryAnalysis-PT dataset (Sun et al., 2025), which provides paired images, tampering masks, and structured textual explanations for forgery detection. This dataset annotates samples from the CASIA2 (Dong et al., 2009b) and MIML (Qu et al., 2024) datasets with multi-level explanatory texts, comprising a total of 15,000 samples. The explanations are organized hierarchically, covering high-level semantic anomalies (*e.g.*, scene inconsistencies), middle-level visual defects (*e.g.*, boundary artifacts), and low-level pixel statistics (*e.g.*, noise mismatches), enabling the model to generate reasoned justifications alongside localization masks.

Fine-tuning was conducted by appending the explanatory texts to the model's output sequences during training. This process encourages TamperTok to produce not only accurate masks but also interpretable rationales, addressing the limitation of forensic expert models that offer limited insight into the reasoning process for end users.

To evaluate the fine-tuned model's explainability, we tested it on three out-of-distribution samples representing distinct tampering types: splicing, copy-move, and AIGC-based forgery. For each sample, the model was prompted with: "Given the image and mask, assess whether the masked region is manipulated and justify under three sections: High-Level Semantic Anomalies; Middle-Level Visual Defects; Low-Level Pixel Statistics." The test images and corresponding model-generated explanations are shown in Figures 6, 7, and 8.

A qualitative analysis of the model's outputs reveals consistent strengths in multi-level reasoning. Across all samples, the explanations begin with a comprehensive summary, highlighting high-level anomalies as primary indicators of doubt, supported by middle- and low-level evidence. For the splicing sample, the model identifies semantic inconsistencies in lighting and physical plausibility, complemented by boundary artifacts and noise mismatches, demonstrating sensitivity to compositing traces. In the copy-move case, it emphasizes intra-scene inconsistencies and textural disruptions, effectively capturing duplication artifacts. For the AIGC sample, the focus on implausible object placements and tonal mismatches underscores the model's ability to detect generative anomalies. Overall, the fine-tuned TamperTok produces structured, evidence-based justifications that align with forensic principles, improving transparency and user trust while maintaining localization accuracy.

## Splice Image

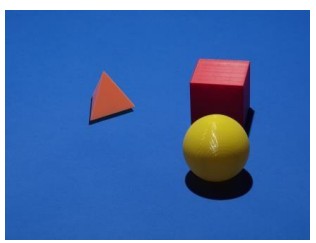

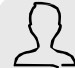 Given the image and mask, assess whether the masked region is manipulated and justify under three sections: High-Level Semantic Anomalies; Middle-Level Visual Defects; Low-Level Pixel Statistics.

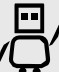

After a comprehensive analysis, we have identified the following clues. High-level anomalies raise serious doubts, while middle-level and low-level findings provide supportive evidence.

**High-Level Semantic Anomalies**

**Scene-wide Lighting Coherence**

The masked triangular object casts a shadow that drifts slightly leftward, whereas the cube and sphere cast strong shadows toward the lower-right. The inconsistent shadow direction suggests the triangle may have been inserted or relit independently.

**Physical Plausibility**

Given the hard, directional light (evident from the sphere's deep shadow), the triangle should show a tight contact shadow along its base. That contact is weak/offset, making the object appear a bit detached from the surface.

**Middle-Level Visual Defects**

**Traces of Tampered Region or Boundary**

Edges of the triangle are uniformly crisp with slight haloing against the blue background. The boundary sharpness differs from edges on nearby objects, typical of cut-and-paste compositing.

**Lighting Inconsistency**

The triangle's two faces exhibit nearly flat, uniform shading and lack a specular highlight, while the cube and sphere show clear highlight/gradient transitions consistent with the strong key light from the upper-right.

**Perspective Relationships**

The triangle's base alignment is subtly at odds with the ground plane implied by other objects and their shadows; its foreshortening does not fully match the shared camera perspective.

**Low-Level Pixel Statistics**

**Noise/Compression Mismatch**

The masked region shows a different grain and micro-contrast than the surrounding scene—either smoother or blockier—indicating it may come from a differently processed source.

**Color and Tone Discrepancies**

The orange hue and highlight tones on the triangle do not blend seamlessly with the global color response (e.g., the warm speculars on the yellow sphere), yielding an abrupt tonal transition at the boundary.

**Textural Differences**

Fine surface texture/banding visible on the cube and sphere is absent or suppressed on the triangle, reinforcing that the masked area is likely edited rather than captured under the same conditions.

**Conclusion:** This image is tampered; I will provide the tampered-region mask.

<|mt_start|><|mt_935|><|mt_238|><|mt_265|><|mt_241|><|mt_893|><|mt_390|><|mt_761|><|mt_739|><|mt_572|><|mt_332|><|mt_300|><|mt_484|><|mt_462|><|mt_781|><|mt_708|><|mt_899|><|mt_481|><|mt_555|><|mt_882|><|mt_918|><|mt_822|><|mt_789|><|mt_776|><|mt_438|><|mt_109|><|mt_908|><|mt_805|><|mt_657|><|mt_397|><|mt_224|><|mt_846|><|mt_171|><|mt_end|>

Figure 6: Test image for splicing tampering type with the model's generated explanation overlaid.

## CopyMove Image

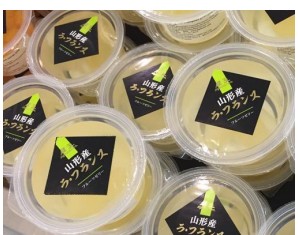

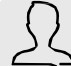 Given the image and mask, `assess` whether the masked region is manipulated and justify under three sections: High-Level Semantic Anomalies; Middle-Level Visual Defects; Low-Level Pixel Statistics.

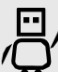

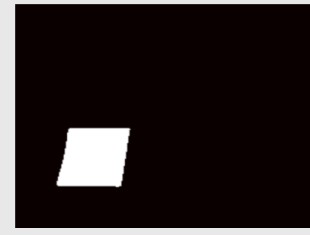

After a thorough examination, we have identified the following clues. High-level anomalies raise significant doubts requiring attention, while middle-level and low-level findings provide solid evidence.

**High-Level Semantic Anomalies**

**Intra-scene Inconsistency:** Among identical dessert lids, the masked label area stands out: its planar patch lacks the curved deformation and concentric highlight seen on neighboring lids. This discrepancy within a set of otherwise uniform items suggests alteration of the label region.

**Middle-Level Visual Defects**

**Traces of Tampered Region or Boundary:** Along the perimeter of the masked quadrilateral, edges appear unnaturally straight and uniformly sharp, with slight haloing against the lid's surface— typical of a pasted patch. The surrounding circular lid textures do not continue seamlessly into the patch.

**Lighting Inconsistency:** Specular highlights and reflections that follow the lid's curvature on adjacent products are interrupted inside the masked area; the patch shows flatter illumination and lacks the radial glare streaks expected on the plastic cover.

**Perspective Relationships:** The patch behaves as a rigid planar insert; its skew does not fully agree with the local curvature and perspective of the round lid, causing a subtle misalignment relative to the ellipse of the rim and neighboring graphics.

**Low-Level Pixel Statistics**

**Noise Inconsistency:** The masked region displays a different grain/compression pattern— either smoother or blockier— than the surrounding lid, indicating it likely originated from a different source or underwent separate processing.

**Color and Tone Mismatch:** Micro-contrast and color tones within the patch differ from adjacent areas; blacks/greys do not blend gradually, and there is an abrupt tonal transition at the boundary.

**Textural Differences:** Fine textures from the lid's concentric ridges and subtle printing dots are muted or absent inside the patch, reinforcing that this area was digitally inserted rather than captured in the original shot.

**Conclusion:** This image is tampered; I will provide the tampered-region mask.

<|mt_start|><|mt_839|><|mt_760|><|mt_389|><|mt_221|><|mt_411|><|mt_875|><|mt_109|><|mt_404|><|mt_749|><|mt_30|><|mt_846|><|mt_520|><|mt_760|><|mt_166|><|mt_363|><|mt_336|><|mt_594|><|mt_224|><|mt_875|><|mt_206|><|mt_604|><|mt_299|><|mt_736|><|mt_122|><|mt_193|><|mt_455|><|mt_594|><|mt_447|><|mt_547|><|mt_162|><|mt_614|><|mt_37|><|mt_end|>

Figure 7: Test image for copy-move tampering type with the model's generated explanation overlaid.

# AIGC Image

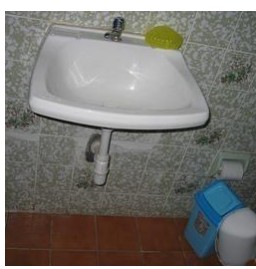

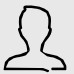 Given the image and mask, assess whether the masked region is manipulated and justify under three sections: High-Level Semantic Anomalies; Middle-Level Visual Defects; Low-Level Pixel Statistics.

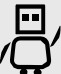

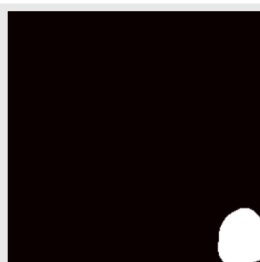

After a thorough examination, we have identified the following clues. High-level anomalies raise significant doubts requiring attention, while middle-level and low-level findings provide solid evidence.

**High-Level Semantic Anomalies**

**Physical Plausibility** Within the masked bottom-right region, the small white cylindrical object's contact with the floor appears implausible. The expected contact shadow on the tile is weak/misaligned, making the object look slightly detached or unnaturally intersecting with the nearby blue bin.

**Middle-Level Visual Defects**

**Traces of Tampered Region or Boundary** The contour of the masked object shows overly crisp, uniform edges with slight haloing, unlike the softer boundaries elsewhere. Tile grout lines that should remain visible exhibit abrupt terminations at the object's edge, suggesting a pasted or re-shaped patch.

**Lighting Inconsistency** Global illumination in the bathroom comes from above/left (seen from the sink's highlights and shadows). Inside the masked area, the white surface is evenly bright with limited shading and an absent/weak cast shadow on the floor, inconsistent with the lighting on adjacent items (blue bin, wall socket).

**Perspective Relationships** The ellipsoidal rim and side curvature of the white cylinder do not fully agree with the floor's perspective (as defined by the converging tile joints). The alignment to the blue bin's top plane also appears slightly off, as if the object was inserted without matching local geometry.

**Low-Level Pixel Statistics**

**Noise Inconsistency** The masked region exhibits a different grain/compression pattern—noticeably smoother than the surrounding tiles and bin—which is typical when content is composited from a different source or processed separately.

**Color and Tone Mismatch** Whites within the masked patch are closer to pure neutral, whereas the rest of the scene shows warmer off-whites due to ambient lighting. This abrupt tonal change at the boundary further indicates editing.

**Textural Differences** Fine texture and micro-contrast seen on adjacent plastic surfaces and floor tiles are muted inside the masked area, reinforcing the conclusion that this portion was digitally altered rather than captured in the original photograph.

**Conclusion:** The image contains manipulated regions. Here is the corresponding mask.

<|mt_start|><|mt_655|><|mt_491|><|mt_103|><|mt_637|><|mt_877|><|mt_711|><|mt_405|><|mt_54|><|mt_833|><|mt_330|><|mt_87|><|mt_454|><|mt_871|><|mt_617|><|mt_917|><|mt_422|><|mt_727|><|mt_593|><|mt_956|><|mt_967|><|mt_996|><|mt_905|><|mt_361|><|mt_382|><|mt_918|><|mt_742|><|mt_933|><|mt_817|><|mt_921|><|mt_293|><|mt_436|><|mt_296|><|mt_end|>

Figure 8: Test image for AIGC tampering type with the model's generated explanation overlaid.

