# OpenReview forum: "TamperTok: Forensics-Driven Tokenized Autoregressive Framework for Image Tampering Localization"
_ICLR.cc/2026/Conference — ICLR 2026 Conference Desk Rejected Submission_

### Official Review · Reviewer_8BYc · 2025-10-29

**Soundness:** 3
**Presentation:** 3
**Contribution:** 3
**Rating:** 8
**Confidence:** 4

**Summary:**

This paper presents TamperTok, a novel framework that reframes Image Manipulation Detection & Localization (IMDL) as an autoregressive sequence generation task. This approach avoids the information bottlenecks of existing MLLM-based methods that rely on external decoders. TamperTok introduces two key components: the Kernel Splatting Decoder (KSD) for stable mask generation from tokens, and Scene-wise Expert Injection (SwEI) to fuse multi-scale forensic features into the MLLM's visual encoder. Extensive experiments show that TamperTok achieves state-of-the-art (SOTA) performance and demonstrates superior robustness on multiple benchmarks.

**Strengths:**

1. The framework introduces a fundamental architectural innovation by reformulating localization as a sequence generation task. This effectively solves the "stitched pipeline" problem common in existing methods, reducing interference from extraneous semantic information and enabling more precise, end-to-end localization.
2. The model demonstrates state-of-the-art performance, significantly outperforming existing forensic and MLLM-based methods across multiple benchmarks. Furthermore, it shows exceptional robustness against common image distortions and strong cross-domain generalization, highlighting its potential for real-world applications.
3. The SwEI module provides a novel and effective method for integrating domain-specific knowledge. By adaptively fusing multi-scale forensic features within the visual backbone, it allows the model to leverage the most relevant clues for different types of manipulations, from subtle splicing artifacts to semantic inconsistencies.

**Weaknesses:**

1. The KSD module is a key innovation of this work, but the paper currently lacks a discussion on the selection of its critical hyperparameters (e.g., the number of clusters for the codebook) and a sensitivity analysis. Adding this would further strengthen the completeness and reproducibility of the module's design.
2. The paper's focus is on methodological innovation and performance validation, where it achieves excellent results. A further discussion on the model's computational efficiency (e.g., parameter count, inference latency) would help readers more comprehensively evaluate its deployment potential in various application scenarios, thus enhancing the practical value of the paper.

**Questions:**

1. The design of the KSD module is convincing, particularly its use of clustering to smooth gradients. We are interested to know how sensitive the model's performance is to key hyperparameters within the KSD, such as the number of clusters for the codebook. An ablation study on this would effectively reinforce the completeness of the module's design.
2. The choice of SparseViT as the expert model for the SwEI module proves effective. Could the authors elaborate on the specific rationale behind this selection?

---

> ### Author Response · Authors · 2025-11-22
> **Response to Reviewer 8BYc (Part 1)**
>
> **General Response**
>
> We sincerely thank the reviewer for the constructive feedback and for acknowledging the potential of our proposed TamperTok framework. We appreciate your detailed questions regarding the hyperparameter sensitivity, computational efficiency, and the rationale behind the expert model selection. We have conducted additional experiments and analyses to address your concerns, which are detailed below.
>
> ---
>
> > **Concern #1 & Question #1: Analysis of KSD clustering hyper-parameters.**
>
> **Response:** We thank the reviewer for suggesting this analysis. To assess the sensitivity of the Kernel Splatting Decoder (KSD) to the clustering hyperparameter $K$ (smoothing_topk), we conducted an ablation study with varying $K$ values.
>
> The results are summarized in the table below:
>
> | $K$ | CASIA1 IoU | CASIA1 F1 | NIST IoU | NIST F1 | Coverage IoU | Coverage F1 | Columbia IoU | Columbia F1 | Glide IoU | Glide F1 | IMD IoU | IMD F1 | Average IoU | Average F1 |
> | :--- | :--- | :--- | :--- | :--- | :--- | :--- | :--- | :--- | :--- | :--- | :--- | :--- | :--- | :--- |
> | $K=1$ (Hard) | 0.75 | 0.87 | 0.38 | 0.56 | 0.57 | 0.74 | 0.88 | 0.88 | 0.65 | 0.80 | 0.62 | 0.75 | 0.64 | 0.77 |
> | $K=5$ | 0.77 | 0.85 | 0.37 | 0.54 | 0.60 | 0.74 | **0.89** | 0.88 | 0.65 | **0.79** | 0.63 | 0.75 | 0.65 | 0.76 |
> | $K=10$ (**Ours**) | **0.79** | **0.87** | **0.43** | **0.60** | **0.61** | **0.79** | **0.89** | **0.90** | **0.66** | 0.78 | **0.65** | **0.78** | **0.67** | **0.79** |
> | $K=20$ | 0.77 | **0.87** | 0.36 | 0.55 | 0.57 | 0.73 | **0.89** | 0.88 | 0.63 | 0.76 | 0.63 | 0.76 | 0.64 | 0.76 |
>
> Setting $K=1$ (hard decoding) is equivalent to the original hard decoding approach used in previous works. We observe that the performance consistently improves as $K$ increases, demonstrating the benefit of our clustering-aware smoothing strategy. While there is a slight drop when $K$ reaches 20, the difference is negligible.
>
> Considering the trade-off between **optimal performance and computational overhead**, we adopted $K=10$ as the default setting. We have explicitly included this ablation study and the corresponding analysis in **Section 4.3.3** of the revised paper.
>
> ---
>
> > **Concern #2: Concerns regarding computational efficiency.**
>
> **Response:** We thank the reviewer for scrutinizing the computational efficiency. To address this concern, we provide a rigorous breakdown of the encoder/decoder parameters and inference time, comparing TamperTok against the MLLM-based method (FakeShield) and the lightweight forensic expert (SparseViT).
>
> The detailed breakdown is shown below:
>
> | Model | Encoder Params | Decoder Params | Total Params | Inference time | Avg F1 |
> | :--- | :--- | :--- | :--- | :--- | :--- |
> | SparseViT | 49.68 M | 0.66 M | ~0.05 B | 0.04 s | 0.76 |
> | FakeShield | 13.35 B (DTE-FDM) | 8.67 B (MFLM) + 641.09 M (SAM) | ~22.66 B | 2.26 s | 0.35 |
> | TamperTok (Ours) | **8.08 B (InternVL) + 99.27 M (SwEI)** | **340.88 M (KSD)** | **~8.52 B** | **1.39 s** | **0.79** |
>
> Based on the updated metrics, we provide a two-fold analysis:
>
> * **Superior Efficiency over MLLM-based Baselines (vs. FakeShield):** TamperTok proves to be significantly more efficient than the existing MLLM-based approach. FakeShield relies on a heavy, cascaded pipeline (running a 13.35B encoder, then an 8.67B decoder, and finally a 641M SAM), which introduces massive computational redundancy. In contrast, TamperTok utilizes a streamlined single-backbone design with a compact decoder (KSD, 340M). This **reduces the total parameter count by ~62%** (8.52B vs. 22.66B) and **inference time by ~38%** (1.39s vs. 2.26s).
> * **Justified Trade-off for Advanced Capabilities (vs. SparseViT):** We acknowledge that TamperTok incurs higher costs than the lightweight SparseViT (0.05B params). However, this trade-off is justified by the critical advantage MLLMs provide over "black-box" experts. Unlike traditional forensic models that output predictions without explanation, the MLLM backbone enables **interpretable reasoning** (e.g., analyzing physical inconsistencies), providing necessary transparency into the decision-making process.
>
> ---

---

> ### Author Response · Authors · 2025-11-22
> **Response to Reviewer 8BYc (Part 2)**
>
> > **Question #2: Rationale for Selecting SparseViT as the Expert Model**
>
> **Response:** We thank the reviewer for this insightful query regarding our design choice.
>
> To identify the optimal expert model, we conducted a comparative study by replacing the default SparseViT with other representative architectures (e.g., MVSS-Net and PSCC-Net) within the SwEI module.
>
> | Method | CASIA1 IoU | CASIA1 F1 | NIST IoU | NIST F1 | Coverage IoU | Coverage F1 | Columbia IoU | Columbia F1 | Glide IoU | Glide F1 | IMD IoU | IMD F1 | Average IoU | Average F1 |
> | :--- | :--- | :--- | :--- | :--- | :--- | :--- | :--- | :--- | :--- | :--- | :--- | :--- | :--- | :--- |
> | *Expert Only:* MVSS-Net | 0.49 | 0.70 | 0.25 | 0.30 | 0.43 | 0.59 | 0.86 | 0.90 | 0.44 | 0.62 | 0.40 | 0.53 | 0.48 | 0.61 |
> | **TamperTok w/ MVSS** | 0.70 | 0.84 | 0.31 | 0.35 | 0.39 | 0.53 | 0.89 | 0.89 | 0.56 | 0.74 | 0.58 | 0.71 | 0.57 | 0.68 |
> | *Expert Only:* PSCC-Net | 0.29 | 0.43 | 0.20 | 0.26 | 0.30 | 0.43 | 0.38 | 0.54 | 0.39 | 0.58 | 0.17 | 0.27 | 0.29 | 0.42 |
> | **TamperTok w/ PSCC** | 0.73 | 0.86 | 0.39 | 0.53 | 0.45 | 0.58 | 0.86 | 0.86 | 0.59 | 0.75 | 0.59 | 0.71 | 0.60 | 0.72 |
> | *Expert Only:* SparseViT | 0.68 | 0.82 | 0.39 | 0.57 | **0.62** | **0.79** | **0.89** | 0.88 | 0.65 | 0.77 | 0.60 | 0.73 | 0.64 | 0.76 |
> | **TamperTok w/ SparseViT** | **0.79** | **0.87** | **0.43** | **0.60** | 0.61 | **0.79** | **0.89** | **0.90** | **0.66** | **0.78** | **0.65** | **0.78** | **0.67** | **0.79** |
>
> The comparison results guided our final selection primarily due to **Superior Absolute Performance:** While our SwEI framework effectively boosts the performance of all tested experts (e.g., improving PSCC-Net from 0.42 to 0.72), SparseViT consistently yields the best overall performance (Average F1 of 0.79) with distinct advantages in handling complex forgery scenarios compared to CNN-based alternatives. Additionally, the marginal **computational cost** difference between these lightweight experts is **negligible** given the dominant load of the MLLM backbone. Consequently, we selected SparseViT as the default expert to maximize detection accuracy without compromising system-level efficiency, and we have updated **Section 4.3.4** to explicitly discuss this rationale.

---

### Official Review · Reviewer_KQPs · 2025-10-29

[review text omitted: it was posted to a different submission]

---

> ### Author Response · Authors · 2025-11-13
>
> Dear Reviewer,
>
> Thank you for taking the time to review our submission. However, it appears that portions of your review may have been inadvertently submitted to the wrong paper, as they seem to reference content or details not present in our work.
>
> Could you please verify this and provide clarification or an update if needed? We appreciate your attention to this matter and look forward to your response.

---

> > ### Comment · Reviewer_KQPs · 2025-11-13
> >
> > Sorry for posting the wrong review comment, and here is the right one.

---

> ### Author Response · Authors · 2025-11-22
> **Response to Reviewer KQPs (Part 1)**
>
> **General Response**
>
> We sincerely thank the reviewer for the positive assessment and for highlighting the novelty and effectiveness of our TamperTok framework. We are encouraged by the reviewer’s recognition of our "novel and elegant" reformulation of the IMDL task and the "comprehensive and rigorous" evaluation. Below, we address the specific concerns regarding the expert module dependency, computational costs, and the choice of expert models.
>
> ---
>
> > **Concern #1: Performance Attribution (Architecture vs. Expert Distillation)**
>
> **Response:** We thank the reviewer for this critical question regarding the source of our performance gains.
>
> We respectfully clarify that while the expert injection contributes to the final performance, the **primary performance leap stems from our architecture**, not merely the expert features. This is explicitly demonstrated in our ablation study (Table 2 in the main paper).
>
> Referring to **Table 2**:
>
> * **Baseline (Row 1):** The vanilla MLLM baseline (LISA-like) achieves an Average F1 of **0.43**.
> * **Architecture Only (Row 2):** By simply introducing our Codebook-based Decoder (CBD) — without any expert injection — the Average F1 jumps significantly to **0.74**. This represents a **0.31 absolute gain**, verifying that the core improvement comes from reformulating the task from a stitched segmentation pipeline to a generative token prediction task.
> * **With Expert (Row 3):** Adding the Scene-wise Expert Injection (SwEI) further improves the F1 from 0.74 to **0.77** (and finally 0.79 with KSD).
>
> The expert module (SwEI) provides a complementary boost (+0.03~0.05 F1), particularly for specific artifacts, but the foundational improvement (+0.31 F1) is driven by the TamperTok architecture itself. Thus, our method is not a simple distillation but a fundamental architectural advancement.
>
> ---
>
> > **Concern #2: Concerns regarding computational costs.**
>
> **Response:** We thank the reviewer for scrutinizing the computational efficiency. To address this concern, we provide a rigorous breakdown of the encoder/decoder parameters and inference time, comparing TamperTok against the MLLM-based method (FakeShield) and the lightweight forensic expert (SparseViT).
>
> The detailed breakdown is shown below:
>
> | Model | Encoder Params | Decoder Params | Total Params | Inference time | Avg F1 |
> | :--- | :--- | :--- | :--- | :--- | :--- |
> | SparseViT | 49.68 M | 0.66 M | ~0.05 B | 0.04 s | 0.76 |
> | FakeShield | 13.35 B (DTE-FDM) | 8.67 B (MFLM) + 641.09 M (SAM) | ~22.66 B | 2.26 s | 0.35 |
> | TamperTok (Ours) | **8.08 B (InternVL) + 99.27 M (SwEI)** | **340.88 M (KSD)** | **~8.52 B** | **1.39 s** | **0.79** |
>
> Based on the updated metrics, we provide a two-fold analysis:
>
> * **Superior Efficiency over MLLM-based Baselines (vs. FakeShield):** TamperTok proves to be significantly more efficient than the existing MLLM-based approach. FakeShield relies on a heavy, cascaded pipeline (running a 13.35B encoder, then an 8.67B decoder, and finally a 641M SAM), which introduces massive computational redundancy. In contrast, TamperTok utilizes a streamlined single-backbone design with a compact decoder (KSD, 340M). This **reduces the total parameter count by ~62%** (8.52B vs. 22.66B) and **inference time by ~38%** (1.39s vs. 2.26s).
> * **Justified Trade-off for Advanced Capabilities (vs. SparseViT):** We acknowledge that TamperTok incurs higher costs than the lightweight SparseViT (0.05B params). However, this trade-off is justified by the critical advantage MLLMs provide over "black-box" experts. Unlike traditional forensic models that output predictions without explanation, the MLLM backbone enables **interpretable reasoning** (e.g., analyzing physical inconsistencies), providing necessary transparency into the decision-making process.
>
> ---

---

> ### Author Response · Authors · 2025-11-22
> **Response to Reviewer KQPs (Part 2)**
>
> > **Question: Sensitivity Analysis of Expert Model Selection**
>
> **Response:** We thank the reviewer for this insightful query regarding the dependency of SwEI on the expert model's capacity.
>
> To address this concern, we replaced the default SparseViT with architecturally different and "weaker" forensic experts (i.e., MVSS-Net and PSCC-Net) within the SwEI module to evaluate their impact.
>
> | Method | CASIA1 IoU | CASIA1 F1 | NIST IoU | NIST F1 | Coverage IoU | Coverage F1 | Columbia IoU | Columbia F1 | Glide IoU | Glide F1 | IMD IoU | IMD F1 | Average IoU | Average F1 |
> | :--- | :--- | :--- | :--- | :--- | :--- | :--- | :--- | :--- | :--- | :--- | :--- | :--- | :--- | :--- |
> | *Expert Only:* MVSS-Net | 0.49 | 0.70 | 0.25 | 0.30 | 0.43 | 0.59 | 0.86 | 0.90 | 0.44 | 0.62 | 0.40 | 0.53 | 0.48 | 0.61 |
> | **TamperTok w/ MVSS** | 0.70 | 0.84 | 0.31 | 0.35 | 0.39 | 0.53 | 0.89 | 0.89 | 0.56 | 0.74 | 0.58 | 0.71 | 0.57 | 0.68 |
> | *Expert Only:* PSCC-Net | 0.29 | 0.43 | 0.20 | 0.26 | 0.30 | 0.43 | 0.38 | 0.54 | 0.39 | 0.58 | 0.17 | 0.27 | 0.29 | 0.42 |
> | **TamperTok w/ PSCC** | 0.73 | 0.86 | 0.39 | 0.53 | 0.45 | 0.58 | 0.86 | 0.86 | 0.59 | 0.75 | 0.59 | 0.71 | 0.60 | 0.72 |
> | *Expert Only:* SparseViT | 0.68 | 0.82 | 0.39 | 0.57 | **0.62** | **0.79** | **0.89** | 0.88 | 0.65 | 0.77 | 0.60 | 0.73 | 0.64 | 0.76 |
> | **TamperTok w/ SparseViT** | **0.79** | **0.87** | **0.43** | **0.60** | 0.61 | **0.79** | **0.89** | **0.90** | **0.66** | **0.78** | **0.65** | **0.78** | **0.67** | **0.79** |
>
> To address your concern regarding the expert dependency, we conducted a comparative study by replacing the default expert in the **SwEI** module with different forensic models.
>
> Our experiments demonstrate that the choice of SparseViT is not strictly crucial for the effectiveness of SwEI; even when substituted with other methods like **MVSS-Net** and **PSCC-Net**, the framework still exhibits competitive performance gains. Specifically, when integrating weaker backbones such as **MVSS-Net** and **PSCC-Net**, the Average F1 scores reached **0.68** and **0.72**, respectively, representing substantial improvements over their standalone baselines of **0.61** and **0.42**. This confirms that the performance boost is primarily driven by the SwEI architecture rather than solely the inherent strength of the expert features.

---

> > ### Comment · Reviewer_KQPs · 2025-11-26
> >
> > Thanks for the authors' response. Currently I have no more questions, and I will keep my score.

---

### Official Review · Reviewer_Vn19 · 2025-10-30

**Soundness:** 4
**Presentation:** 3
**Contribution:** 4
**Rating:** 8
**Confidence:** 5

**Summary:**

This paper introduces a novel Multimodal Large Language Model (MLLM) framework named TamperTok for Image Manipulation Detection and Localization (IMDL). The authors astutely reframe the localization task as an autoregressive token generation problem, thereby creating an end-to-end trainable system that avoids the pitfalls of prior methods relying on separate, non-integrated segmentation decoders. The core contributions are twofold: the Kernel Splatting Decoder (KSD), which enables stable, direct supervision for mask generation by smoothing token gradients, and the Scene-wise Expert Injection (SwEI) module, which intelligently fuses multi-scale forensic features from an expert model into the MLLM's visual backbone. Comprehensive experimental results demonstrate that TamperTok not only sets a new state-of-the-art across numerous benchmarks but also exhibits impressive robustness and cross-domain generalization.

**Strengths:**

1. Elegant End-to-End Formulation: The paper's primary strength lies in its reformulation of a localization problem into a sequence generation task. This is an elegant solution that directly addresses the information bottleneck and semantic mismatch issues that plague existing MLLM-based methods. By making the entire pipeline from image to mask tokens directly supervisable and end-to-end differentiable, the framework fully leverages the MLLM's spatial reasoning capabilities.
2. Rigorous and Meaningful Evaluation: The experimental design is exceptionally thorough. The authors go beyond standard in-domain tests and place a strong emphasis on cross-domain generalization by evaluating on six diverse, unseen datasets. This is a critical and often overlooked aspect in the forensics field, and the model's strong performance in these challenging scenarios convincingly demonstrates its real-world viability.
3. Sophisticated Knowledge Integration: The Scene-wise Expert Injection (SwEI) module is a well-designed and highly effective method for integrating domain-specific knowledge. Unlike naive late-fusion approaches that can dilute important signals, SwEI's attention-based, multi-scale injection at intermediate stages of the visual encoder ensures that subtle forensic artifacts are preserved and effectively aligned with the visual features. This sophisticated fusion strategy is a key reason for the model's high performance, especially in detecting challenging manipulations like copy-move.

**Weaknesses:**

1. Regarding the SwEI module, the paper mentions that it can fuse features at different levels to handle various types of tampering . However, the paper does not further explain whether the strength of the injected expert model features changes when facing different tampering types (e.g., splicing, copy-move).
2. The paper does not provide an ablation study for the clustering hyper-parameters used in the KSD module's loss calculation . For instance, no justification is given for the choice of the number of code-book clusters, making it difficult for readers to fully understand the design decisions of the module and its sensitivity to the final performance.
3. Some references have wrong information or format. For example, the reference in lines 560-562 should be [1].

[1] Chenfan Qu, Yiwu Zhong, Chongyu Liu, et al. Towards modern image manipulation localization: A large-scale dataset and novel methods[C]//Proceedings of the IEEE/CVF Conference on Computer Vision and Pattern Recognition. 2024: 10781-10790.

**Questions:**

The authors are encouraged to address the points raised in the 'Weaknesses' section above.

---

> ### Author Response · Authors · 2025-11-22
> **Response to Reviewer Vn19 (Part 1)**
>
> **General Response**
>
> We sincerely appreciate your positive assessment and your recognition of our end-to-end formulation and rigorous cross-domain evaluation. We have addressed your concerns below.
>
> ---
>
> > **Concern #1: Analysis of injected expert feature strength across different tampering types.**
>
> **Response:** We thank the reviewer for this insightful comment regarding the adaptability of our **SwEI** module.
>
> To quantify the contribution of the expert features, we define the Average Relative Expert Intensity ($I_{avg}$). It is defined as:
>
> $$
> I_{avg} = \frac{1}{B} \sum_{i=1}^{B} \frac{RMS(F^{sparse}_i)}{RMS(F^{vit}_i)}
> $$
>
> $$
> RMS(X) = \sqrt{\frac{1}{C \cdot H \cdot W} \sum_{c,h,w} X_{(c,h,w)}^2}
> $$
>
> $$
> F_{sparse} = \alpha \cdot \text{Proj}(E_s) + \beta \cdot \text{Proj}(E_d)
> $$
>
> where:
> * $B$: The number of samples in the evaluation set.
> * $RMS(X)$: The Root Mean Square function calculating the quadratic mean magnitude over the channel ($C$), height ($H$), and width ($W$) dimensions.
> * $F_{vit}$: The intermediate feature map extracted from the MLLM backbone (InternVL) at the corresponding layer.
> * $F_{sparse}$: The adaptively fused expert feature map derived from the Scene-wise Expert Injection (SwEI) module.
> * $E_s$ and $E_d$: $E_s$ is the shallow feature and $E_d$ is deeper feature. The multi-scale feature maps with complementary characteristics (e.g., texture-rich and semantic-rich) from the expert model (SparseViT).
> * $\text{Proj}(\cdot)$: The projection layer that aligns the channel dimensions of expert features with the backbone.
> * $\alpha$, $\beta$: The dynamic attention map generated by the backbone features, used to adaptively weigh the importance of different expert scales.
>
> We have included the detailed experimental results and visualization in the newly added **Appendix A.4**. In addition to the radar chart presented in the appendix, we provide the specific numerical values of $I_{avg}$ across different tampering types in the table below to clearly illustrate the contribution of expert features at different layers:
>
> | Tampering Type | Intermediate Layers ($I_{avg}$) | Deep Layers ($I_{avg}$) |
> | :--- | :--- | :--- |
> | AIGC-Editing | **1.982** | 1.082 |
> | Splice | 1.91 | 1.061 |
> | Copy-Move | 1.914 | 1.136 |
> | Removal | 1.933 | **1.211** |
>
> The data illustrates that **SwEI** acts as a dynamic, adaptive compensator rather than a static addition:
>
> * **AIGC-Editing (Texture-Driven):** The expert intensity peaks at **1.982** in the Intermediate Layers. This confirms that the model actively prioritizes expert injection to capture subtle low-level artifacts, such as diffusion fingerprints. It effectively compensates for the MLLM backbone, which typically lacks sensitivity to high-frequency details.
> * **Removal (Semantic-Driven):** We observe the highest Deep Layer intensity (**1.211**) for removal operations. This indicates a necessity for expert guidance to identify "inpainted" areas. Although these regions often fit the semantic context, they introduce structural inconsistencies that the backbone alone may fail to detect.
> * **Splicing & Copy-Move:** **Splicing** exhibits the lowest deep-layer intensity (1.061), suggesting the MLLM backbone is sufficient for resolving blatant context contradictions like object mismatches. Conversely, **Copy-Move** requires higher expert engagement (1.136) in the deep layers, utilizing structural cues to detect repeated internal patterns.
>
> ---

---

> ### Author Response · Authors · 2025-11-22
> **Response to Reviewer Vn19 (Part 2)**
>
> > **Concern #2: Lack of ablation study for KSD clustering hyper-parameters.**
>
> **Response:** We thank the reviewer for raising this important question about hyperparameter sensitivity.
>
> We conducted an ablation study on the hyperparameter $K$ (smoothing_topk), which controls the number of nearest codes used for label smoothing. The table below details the IoU and F1 scores across all six datasets with varying $K$:
>
> | $K$ | CASIA1 IoU | CASIA1 F1 | NIST IoU | NIST F1 | Coverage IoU | Coverage F1 | Columbia IoU | Columbia F1 | Glide IoU | Glide F1 | IMD IoU | IMD F1 | Average IoU | Average F1 |
> | :--- | :--- | :--- | :--- | :--- | :--- | :--- | :--- | :--- | :--- | :--- | :--- | :--- | :--- | :--- |
> | $K=1$ (Hard) | 0.75 | 0.87 | 0.38 | 0.56 | 0.57 | 0.74 | 0.88 | 0.88 | 0.65 | 0.80 | 0.62 | 0.75 | 0.64 | 0.77 |
> | $K=5$ | 0.77 | 0.85 | 0.37 | 0.54 | 0.60 | 0.74 | **0.89** | 0.88 | 0.65 | **0.79** | 0.63 | 0.75 | 0.65 | 0.76 |
> | $K=10$ (**Ours**) | **0.79** | **0.87** | **0.43** | **0.60** | **0.61** | **0.79** | **0.89** | **0.90** | **0.66** | 0.78 | **0.65** | **0.78** | **0.67** | **0.79** |
> | $K=20$ | 0.77 | **0.87** | 0.36 | 0.55 | 0.57 | 0.73 | **0.89** | 0.88 | 0.63 | 0.76 | 0.63 | 0.76 | 0.64 | 0.76 |
>
> Setting $K=1$ (hard decoding) is equivalent to the original hard decoding approach used in previous works. We observe that the performance consistently improves as $K$ increases, demonstrating the benefit of our clustering-aware smoothing strategy. While there is a slight drop when $K$ reaches 20, the difference is negligible.
>
> Considering the trade-off between **optimal performance and computational overhead**, we adopted $K=10$ as the default setting. We have explicitly included this ablation study and the corresponding analysis in **Section 4.3.3** of the revised paper.
>
> ---
>
> > **Concern #3: Incorrect reference information.**
>
> **Response:** We thank the reviewer for carefully checking our references. We **have corrected the reference** for the MIML dataset to: "Chenfan Qu, Yiwu Zhong, Chongyu Liu, et al. Towards Modern Image Manipulation Localization: A Large-Scale Dataset and Novel Benchmark Methods. In CVPR, 2024, pp. 10781-10790."
>
> We have also proofread the entire bibliography to ensure accuracy.

---

### Official Review · Reviewer_Zn9c · 2025-11-01

**Soundness:** 4
**Presentation:** 4
**Contribution:** 4
**Rating:** 8
**Confidence:** 4

**Summary:**

TamperTok reframes image tampering localization (IMDL) as an autoregressive token generation task using multimodal LLMs.
Instead of relying on external segmentation decoders, it introduces the Kernel Splatting Decoder (KSD) for direct spatial mask prediction and the Scene-wise Expert Injection (SwEI) module to inject multi-scale forensic features into the visual encoder.
Across six benchmarks, TamperTok outperforms both prior MLLM-based methods (FakeShield, SIDA) and expert forensic models (SparseViT), improving IoU/F1 by around 20% and showing strong robustness to compression, noise, and scaling. The appendix also shows promising explainability results with structured textual reasoning.

**Strengths:**

- TamperTok is the first fully end-to-end autoregressive framework for image tampering localization using multimodal LLMs.
- It reformulates tampering detection as a token generation task, eliminating the need for an external segmentation head. This has benchmark-level significance for future multimodal tampering detection research and provides a strong foundation for further extensions.
- The design and reasoning around the codebook similarity avoidance in the proposed KSD module are very sound. In particular, the mechanism to prevent gradient oscillation caused by similar codebook entries, and the methods to mitigate sharp gradients, appear novel both in theory and in practical implementation.
- Extensive experiments demonstrate the solid performance of the proposed model.

**Weaknesses:**

- The backbone choice (internVL in this paper) of the large model is quite important — it’s better to include this information in the main text to make the paper more self-contained, rather than mentioning it only in the appendix.
- It would be helpful if the paper could clarify how the codebook is selected or trained, including the data and procedure used for its construction, which would make the methodology more transparent and reproducible.

**Questions:**

Please refer to the weakness section.

---

> ### Author Response · Authors · 2025-11-22
> **Response to Reviewer Zn9c**
>
> **General Response**
>
> We thank the reviewer for the positive assessment and for recognizing the novelty of TamperTok framework. We have addressed the comments regarding the backbone choice and codebook transparency in the revised paper.
>
> ---
>
> > **Concern #1: The backbone choice (InternVL) should be included in the main text.**
>
> **Response:** We thank the reviewer for pointing out this omission. We agree that the backbone information is critical for the paper's self-containment. We have revised **Section 3.1** to explicitly state that TamperTok is constructed based on **InternVL2.5-8B**.
>
> ---
>
> > **Concern #2: Clarification on codebook selection and training.**
>
> **Response:** We thank the reviewer for this opportunity to clarify the details of our codebook. In the revised **Appendix A.2**, we have explicitly stated that we utilize the pre-trained codebook weights from **HiMTok**.
>
> To ensure self-containment, we provide the specific details of how this codebook was obtained:
>
> * **Codebook Construction:** The codebook comprises a vocabulary of 1,024 learnable latent vectors. The architecture employs a ViT-based Encoder to compress a 2D binary mask into a compact 1D latent sequence (e.g., 32 tokens). During the forward pass, these continuous latent vectors are quantized by mapping them to their nearest neighbors in the codebook. Subsequently, a ViT-based Decoder reconstructs the original 2D mask from this discrete 1D token sequence.
>
> * **Training Data:** The codebook was trained on a large-scale composite segmentation dataset, including ADE20K, COCO, etc., to ensure robust generalization.
>
> * **Loss Functions:** To adapt the codebook to tampering localization, we fine-tune it using the **Hierarchical Mask Loss** (HML) alongside our proposed **Kernel Splatting Decoder loss** ($L_{KSD}$). HML enforces a coarse-to-fine generation hierarchy by supervising tokens with progressively sharper Gaussian-blurred masks. Complementarily, $L_{KSD}$ mitigates the gradient sharpness of hard quantization via clustering-aware label smoothing, ensuring stable end-to-end optimization.

---

> > ### Comment · Reviewer_Zn9c · 2025-11-27
> > **Good paper, clear Accept**
> >
> > Thanks for the author's response, which solved my concerns. I believe this paper holds important research value for future research in image tampering localization. I keep my score of "accept, good paper".

---

### Author Response · Authors · 2025-12-02
**Summary of Rebuttal and Discussions**

We acknowledge the recent notification from the Program Chairs regarding the anonymity breach on OpenReview. **We fully extend our sincere gratitude to the Area Chairs, Reviewers, and the Program Committee for their dedication and professionalism under these exceptional circumstances.**

We formally summarize the core technical points addressed in our rebuttal below to ensure the merits of our work are clearly communicated.

### Summary of Key Responses to Reviewer Concerns

> **1. Adaptability & Robustness of SwEI Module (Reviewers Vn19, KQPs & 8BYc)**

* **Sensitivity to Expert Selection:** Substitution experiments utilizing alternatives like MVSS-Net and PSCC-Net confirm the framework **consistently amplifies standalone capabilities**. For instance, our method yields a **+0.3 improvement in F1 score** over PSCC-Net alone. This validates the **architecture-driven** nature of the improvements, supporting the final selection of SparseViT for its superior absolute performance.
* **Adaptability to Tampering Types:** We analyzed the Average Relative Expert Intensity ($I_{avg}$), calculated as the RMS ratio of expert features to backbone representations, which reveals that SwEI adjusts its **layer-wise activation** based on the tampering category. Intermediate layers show stronger response for texture anomalies like AIGC-Editing. In contrast, the focus shifts to deep layers for semantic inconsistencies such as Removal.

> **2. KSD Hyperparameters & Codebook Details (Reviewers Zn9c, Vn19 & 8BYc)**

* **Ablation on Smoothing Hyperparameters:** We conducted a sensitivity analysis on the clustering hyperparameter $K$. Results show consistent improvements over hard decoding ($K=1$). We adopted **$K=10$** as the default setting to **balance optimal performance against computational overhead**, noting that performance gains saturate beyond this point.
* **Implementation Details of Codebook:** We clarified that we utilize a pre-trained codebook (1,024 vocabulary) from HiMTok, fine-tuned with Hierarchical Mask Loss and our proposed KSD loss to ensure robust generalization.

> **3. Computational Efficiency (Reviewers KQPs & 8BYc)**

* We provided a breakdown showing TamperTok **reduces parameter count by 62%** (8.52B vs. 22.66B) and **inference time by 38%** (1.39s vs. 2.26s) compared to the MLLM-based baseline (FakeShield).
* Notably, this streamlined architecture achieves superior accuracy, surpassing the baseline with a **0.33 improvement in F1 score** (0.46 vs. 0.79).

---

### Summary of Discussions

We extend our sincere gratitude to the Area Chairs and Reviewers for their time and dedicated efforts throughout the review process. During the discussion, we are particularly encouraged by the recognition of our work, which was highlighted as a **“fundamental architectural innovation”** by **Reviewer 8BYc**, an **“elegant solution”** by **Reviewer Vn19**, and for its **“comprehensive and rigorous”** evaluation by **Reviewer KQPs**. Additionally, **Reviewer Zn9c** noted its **“benchmark-level significance”** for future multimodal tampering detection research.

Briefly, TamperTok reformulates IMDL as a mask token generation task, incorporating the Kernel Splatting Decoder (KSD) and Scene-wise Expert Injection (SwEI) to achieve SOTA performance. During the rebuttal, reviewers provided valuable feedback regarding hyperparameter sensitivity, expert adaptability, and computational efficiency. **We addressed these queries through extensive ablation studies and comparative efficiency breakdowns.**

These results explicitly validate TamperTok’s core strengths:
* TamperTok introduces a novel **mask token generation paradigm** to mitigate the reliance on the extra exogenous segmentation decoder.
* TamperTok achieves new SOTA with **0.33 improvement in F1 score** (0.46 vs. 0.79) and **0.23 improvement in IoU** (0.44 vs. 0.67), while simultaneously reducing both **38% computational cost** (1.39s vs. 2.26s) and **62% model parameters** (8.52B vs. 22.66B) compared to FakeShield.

Following our responses, **Reviewers Zn9c and KQPs explicitly confirmed that their concerns were resolved and maintained their original positive scores.**

Finally, we thank the AC and Reviewers again for their constructive contributions to improving this paper. We hope this summary assists in the final assessment of our work.

---

### Note · Program_Chairs · 2025-12-08
**Submission Desk Rejected by Program Chairs**

Hallucinated references are against ICLR policies:
"Chong Zou, Zhipeng Wang, Ziyu Li, Nan Wu, Yuling Cai, Shan Shi, Jiawei Wei, Xia Sun, Jian
Wang, and Yizhou Wang. Segment everything everywhere all at once. In Advances in Neural
Information Processing Systems (NeurIPS), volume 36, 2023."